# Prevalence of Aleutian Mink Disease Virus (AMDV) in Free-Ranging American Mink from Biebrza and Narew National Parks (Poland)—An Epidemiological Concern

**DOI:** 10.3390/ani14172584

**Published:** 2024-09-05

**Authors:** Konrad Przywara, Jan Siemionek, Tadeusz Jakubowski, Klaudia Konczyk-Kmiecik, Anna Szczerba-Turek

**Affiliations:** 1Veterinarian Konrad Przywara, 40A Grudzielskiego St., 63-700 Krotoszyn, Poland; k.przywara@o2.pl; 2Department of Epizootiology, Faculty of Veterinary Medicine, University of Warmia and Mazury, Oczapowskiego 13, 10-718 Olsztyn, Poland; jan.siemionek@uwm.edu.pl (J.S.); klaudia.konczyk@uwm.edu.pl (K.K.-K.); 3Department of Large Animal Diseases with Clinic, Faculty of Veterinary Medicine, Warsaw University of Life Sciences, 02-797 Warsaw, Poland; kruk02@o2.pl; 4Laboratory of the Polish Society of Breeders and Producers of Fur Animals, Pocztowa St. 5, 62-080 Tarnowo Podgórne, Poland

**Keywords:** Aleutian disease, plasmacytosis, AMDV, *NS1*, free-ranging American mink, phylogenetic analysis, epidemiological concern, Narew National Park, Biebrza National Park

## Abstract

**Simple Summary:**

Aleutian Mink Disease Virus (AMDV) causes Aleutian disease, a chronic disorder affecting both farmed and free-ranging mink. This study aimed to analyze AMDV variants from free-ranging American mink in northeastern Poland. Between 2018 and 2019, 26 spleen samples were collected from minks in Narew National Park (NNP) and Biebrza National Park (BNP). DNA was extracted and subjected to PCR to amplify the *NS1* gene, followed by sequencing and phylogenetic analysis. The *NS1* gene was detected in 50% of samples from NNP minks and in 30% of samples from BNP minks, with an overall prevalence of 42.31%. Phylogenetic analysis revealed high homology among most sequences, indicating a stable, potentially endemic AMDV population, although unique variants suggest ongoing genetic diversity. These findings highlight the significant prevalence of AMDV in free-ranging mink and underscore the need for continued monitoring and research to understand the virus’s transmission dynamics and impact on wildlife and mink farming.

**Abstract:**

Aleutian Mink Disease Virus (AMDV) is the causative agent of Aleutian disease (AD). This progressive and chronic disorder significantly impacts the mink breeding industry, affecting farmed and free-ranging American and European mink. This study investigated AMDV variants isolated from free-ranging American mink in northeastern Poland. Between 2018 and 2019, 26 spleen samples were collected from mink in Narew National Park (NNP) and Biebrza National Park (BNP). DNA was extracted and subjected to PCR to amplify the *NS1* gene, followed by sequencing and phylogenetic analysis. The *NS1* gene was detected in 50% of samples from NNP minks and in 30% of samples from BNP minks, with an overall prevalence of 42.31%; these findings align with global data and indicate serious ecological and health concerns. Ten closely related AMDV variants and one distinct variant were identified. The grouped variants exhibited high genetic homogeneity, closely related to strains found in mink from the USA, Germany, Greece, Latvia, and Poland; meanwhile, the distinct variant showed similarities to strains found in mink from Finland, Denmark, China, Poland, and Latvia, suggesting multiple infection sources. These findings, consistent with data from Polish mink farms, indicate significant genetic similarity between farmed and wild mink strains, suggesting potential bidirectional transmission. This underscores the importance of a One Health approach, emphasizing the interconnectedness of human, animal, and environmental health. Continuous surveillance and genetic studies are crucial for understanding AMDV dynamics and mitigating their impacts. Measures to reduce transmission between farmed and wild mink populations are vital for maintaining mink health and ecosystem stability.

## 1. Introduction

Aleutian Mink Disease Virus (AMDV) is the causative agent of Aleutian disease (AD) or mink plasmacytosis, a progressive and chronic disorder affecting the mink breeding industry [1,2,3]. This viral disease predominantly affects farmed and free-ranging American mink (*Neovison vison*) and free-ranging European mink (*Mustela lutreola*) [4]. However, it has also been detected in other members of the family Mustelidae, such as martens (*Martes* spp.) and short-tailed weasels (*Mustela erminea*), and in other families: Felidae—lynx (*Lynx* spp.); Canidae—arctic fox (*Vulpes lagopus*), red fox (*Vulpes vulpes*), and raccoon dog (*Nyctereutes procyonoides*); Mephitidae—striped skunk (*Mephitis mephitis*); Viverridae—genet (*Genetta genetta*) [4,5,6,7,8,9,10,11]. The International Committee on Taxonomy of Viruses defined AMDV as the *Carnivore amdoparvovirus* 1 species, as a member of the genus *Amdoparvovirus*, within the family *Parvoviridae* and subfamily *Parvovirinae* [12,13]. The AMDV is a non-enveloped, single-stranded DNA genome with a size of 4.5–5.0 kilobase (kb) [14,15]. The genome of the virus contains two structural proteins, VP1 and VP2, as well as three non-structural proteins, NS1, NS2, and NS3 [1,16,17]. NS1–NS3 are produced through alternative splicing mechanisms and play crucial roles in regulating the late synthesis of structural proteins and in the replication of the virus’s nucleic acids [17,18,19]. The capsid is primarily composed of the structural protein VP2 along with the smaller protein VP1, both serving as the main immunogenic antigen proteins of AMDV [17,18]. Sequence variations in the genes encoding NS1 and VP2 proteins are commonly utilized in phylogenetic analyses and epidemiological investigations [20,21,22,23,24,25].

AMDV infection outcomes vary depending on factors such as the strain and dose of the virus, the age of the host, and the host genotype [12,26]. AD can cause acute and chronic symptoms with subclinical or asymptomatic presentations [1]. Chronic infections typically result in hair depigmentation, weakness, weight loss, decreased appetite, and diarrhea [1,27]. The subclinical course is marked by reproductive issues like reduced pregnancies, smaller litters, and spontaneous terminations [28,29,30]. In clinical cases, AMDV induces a progressive syndrome in adult mink known as chronic AD, characterized by immune-complex deposition, weight loss, hypergammaglobulinemia, and necrotizing arteritis [1,8,27,31].

In mink, the primary mode of virus transmission is typically vertical, passing from infected females to their offspring [32]. However, horizontal transmission through blood, feces, urine, and saliva, is also possible [3,33,34]. The AMDV can contaminate cage surfaces, gloves, protective footwear, or even the tracks left by vehicle tires [34,35,36]. Notably, free-ranging mink can serve as reservoirs for the virus, thus endangering farmed mink [3,34,37,38]. Given the virus’s longevity in the environment, mink farms can experience outbreaks even after the apparent eradication of the disease. No effective treatment or vaccine exists, and outbreaks lead to substantial economic losses for mink farms [2,39]. The main problems with eradication programs seem to be the following: first, persistence; second, the high concentration of farms; third, high seroprevalence. None of the programs achieved satisfactory results [2,35,38,40,41]. 

Poland ranks among the top mink farming nations globally, which underscores the significant risk of transmission between farmed and free-ranging populations. As per the Act of 11 March 2004 on the protection of animal health and eradication of infectious animal diseases, AD is subject to registration in Poland [42]. AMDV is a subject of interest among Polish scientists [20,23,24,29,31,37,43,44]. According to the latest investigation into farmed mink in Poland, two separate clads of AMDV variants have been identified, suggesting that two independent pools of viruses persist in farms [20,24]. The first clad (group A) consists of AMDV variants that show approximately 95% similarity with the nonpathogenic strain AMDV-G (Acc. No. NC001662); meanwhile, the second clad (group B) consists of AMDV variants that show approximately 86% similarity with the nonpathogenic strain AMDV-G (Acc. No. NC001662). Variants from the first clad were isolated from farmed mink located in the West Pomeranian Voivodeship in northwestern Poland [20,24]. AMDV variants from the second clad persist in mink farms located in Greater Poland, Podkarpackie, Lesser Poland, Podlaskie, and Pomeranian Voivodeships. AMDV variants from the first clad indicated high homogeneity in the viral pool, while the AMDV variants from the second clad showed less homogeneity in the viral pool, which may be recognized in some sub-clads [20,24]. However, the data on the prevalence of AMDV in farmed and free-ranging mink in Poland seem insufficient from an epidemiological perspective, and further research is warranted. In line with the “Spill-over and Spill-back” strategy, free-ranging mink with AMDV infections can jeopardize farmed mink upon contact, potentially introducing the virus to a susceptible farm population [34,45]. Conversely, infected farmed mink might transmit the virus to free-ranging populations, ensuring its environmental persistence. Despite the American mink being recognized as an invasive species in Europe, concerns linger regarding the disease’s ramifications on free-ranging populations and the encompassing ecosystem [34,46,47].

The probable prevalence of AMDV in free-ranging American mink in Poland emerges as a significant epidemiological challenge, considering Poland’s pivotal role in mink farming and the potential interchange of the virus between free-ranging and farmed populations. Consistent monitoring, in-depth research, and rigorous biosecurity protocols are paramount in managing and alleviating the disease’s impacts. The primary aim of this study was to analyze AMDV variants isolated from free-ranging American mink from northeastern Poland (Biebrza and Narew National Parks).

## 2. Materials and Methods

### 2.1. Research Material

Between 2018 and 2019, we collected 16 spleen samples from free-ranging American mink inhabiting Narew National Park (NNP) (latitude 53.2492° N, longitude 23.4700° E) and 10 spleen samples from mink within Biebrza National Park (BNP) (latitude 53.5° N, longitude 22.8° E). The mink trapping was part of conservation efforts under the LIFE+ project “Protection of water and marsh birds in five National Parks” (LIFE09 NAT/PL/000263). This project was implemented to restore habitats and manage invasive species, such as the American mink, threatening native species in these protected areas [48]. The trapping and euthanasia procedures were conducted in compliance with national and local regulations, specifically under the oversight of the Ministry of Climate and Environment of Poland and the relevant park authorities, including the Directorate for Environmental Protection and the Veterinary Inspection Services. These bodies ensured that all animal welfare regulations were followed, with approval from the Local Ethical Commission for Animal Experiments in Białystok. The euthanasia process was carried out by trained personnel using CO_2_, following ethical guidelines aiming to minimizing suffering. This approach ensured that all practices adhered to the highest standards of humane treatment of wildlife, as required by Polish law.

### 2.2. DNA Extraction

Total DNA was isolated from 10 mg spleens using a Genomic Mini DNA isolation kit (A&A Biotechnology, Gdynia, Poland) following the manufacturer’s protocol. DNA was eluted in 150 µL of elution buffer. The quality and quantity of the extracted DNA were assessed using a BioPhotometer^®^ (Eppendorf, Hamburg, Germany). Extracted DNA was stored at −20 °C for future molecular biology experiments.

### 2.3. PCR

PCR was conducted for all DNA samples extracted from spleens using primers amplifying a fragment of the *NS1* gene (AMDV-F-7-H-PN1: 5′ CATATTCACTGTTGCTTAGGTTA 3′ and AMDV-R-7-HPN2: 5′ CGTTCTTT-GTTAGTTAGGTTGTC 3′), as previously described by Jensen [49]. The reaction mixture contained 120 ng DNA (3 µL), 12.5 µL JumpStartTM REDTaq^®^ ReadyMixTM Reaction Mix (Sigma-Aldrich, St. Louis, MO, USA), approximately 0.1 µL of each primer (AMDV-F-7-H-PN1 and AMDV-R-7-HPN2)—the final concentration of each primer was 0.4 µM—and 9.3 µL water. The total volume was 25 µL. The reaction conditions were as follows: initial denaturation—94 °C for 2 min; thirty-four cycles of 94 °C for 30 s, 55 °C for 30 s, and 72 °C for 2 min; final extinction at 72 °C for 5 min in a Mastercycler (Eppendorf, Hamburg, Germany). The reaction products were separated on a 2% agarose gel with Midori Green Advance DNA stain (NIPPON Genetics EUROPE, Düren, Germany) at 80 V. Visualization and archiving of the gel were conducted using the Gel Doc Gel Documentation System (BioRad, Hercules, CA, USA).

### 2.4. Sequencing and Phylogenetic Analysis

The amplicons were cleaned using a clean-up kit (A&A Biotechnology, Gdynia, Poland) according to the manufacturer’s instructions. The purified amplicons were sequenced at Genomed S. A. (Warsaw, Poland). The sequence data were compared with the nucleotide sequence of other AMDVs from NCBI using BLAST+ 2.15.0 [50]. The sequence analysis was conducted using BioEdit Sequence Alignment Editor, Computational Evolutionary Biology package MEGAX, and Clustal W software [51,52,53]. The phylogenetic and evolutionary history analyses were performed using the maximum composite likelihood method [51,54]. The sequences used in the phylogenetic analysis are listed in Table 1. 

### 2.5. Statistical Analysis

The prevalence of the *NS1* gene of AMDV strains in free-ranging American mink populations was evaluated using the Clopper–Pearson ‘exact’ method, which relies on the beta distribution at a significance level of α = 0.05. Statistical analyses were performed using EpiTools, a set of free epidemiological calculators, to conduct comparisons [55]. 

**Table 1 animals-14-02584-t001:** Other known sequences of *NS1* gene of AMDV isolates used in this study (note: ND—no data).

Accession Number (Acc. No.)	Strain	Virulence	Country	Type	References
NC 001662	AMDV G	Nonpathogenic	USA	Farmed	[14]
X77083	AMDV-Utah	Highly virulent	USA	Farmed	[56]
X77084	AMDV-United	Highly virulent	Denmark	Farmed	[56]
X77085	AMDV-K	Highly virulent	Denmark	Farmed	[56]
X97629	AMDV-SL3	Intermediately virulent	Germany	Farmed	[57]
MN175700	AMDV ZP1b (I)	ND	Poland	Farmed	[20]
MN175735	AMDV LUB1 (II)	ND	Poland	Farmed	[20]
MN175724	AMDV WLKP 2a (III)	ND	Poland	Farmed	[20]
MN175729	AMDV PODK2a (IV)	ND	Poland	Farmed	[20]
KY997051	AMDV_mink-f_PL_459	ND	Poland	Farmed	[22]
OR123963	AMDV A1	ND	Latvia	Farmed	[24]
OR123971	AMDV B4	ND	Latvia	Farmed	[24]
MG366646	AMDV F8B5	ND	Greece	Farmed	[22]
GU183265	AMDV ADV-LN2	Highly virulent	China	Farmed	[58]
MG821256	AMDV HY 47	ND	Finland	Wild	[59]
MG821257	AMDV HY 51	ND	Finland	Wild	[59]
MG821282	FIN17/F40	ND	Finland	Farmed	[59]
OK263135	AMDV BPN5	ND	Poland	Wild	This study
OK263139	AMDV NPN4	ND	Poland	Wild	This study
OK263140	AMDV NPN5	ND	Poland	Wild	This study
OK263142	AMDV NPN7	ND	Poland	Wild	This study

## 3. Results

### 3.1. Prevalence of AMDV in Free-Ranging American Mink

The presence of the *NS1* gene of AMDV was confirmed in 8 out of 16 spleen samples (50%, 95% CI = 24.65–75.35) taken from free-ranging American mink in NNP, and in 3 out of 10 spleen samples (30%, 95% CI = 6.67–65.25) from those in BNP. The overall prevalence of AMDV in free-ranging American mink across both national parks was 42.31% (11 out of 26, 95% CI = 23.35–63.08). There is no statistically significant difference in the prevalence of AMDV in American mink between the two examined national parks. 

### 3.2. Analysis of NS1 Gene of AMDV

The sequences obtained during this study had lengths of 374 base pairs (bp); they were deposited in the GenBank databases at the National Center for Biotechnology Information (NCBI) with the following Accession Numbers (Acc. No.): OK263135–OK263145. The evolutionary relationship between the *NS1* gene variants obtained in this study and AMDV G (Acc. No. NC001662) is presented in Figure 1. 

After the phylogenetic analyses, the isolates divided into one group with 10 closely related isolates and one group with separate isolates: OK263142—AMDV NPN7 Wild PL. In the group, seven isolates (70.00%, 95% CI = 34.75–93.33) (Acc. No.: OK263135—AMDV BPN5 Wild PL; OK263136—AMDV BPN7 Wild PL; OK263137—AMDV BPN8 Wild PL; OK263138—AMDV NPN3 Wild PL; OK263141—AMDV NPN6 Wild PL; OK263144—AMDV NPN9 Wild PL; OK263145—AMDV NPN16 Wild PL) showed 100% homology in the nucleotide sequences of the *NS1* gene. For further analysis, we selected OK263135 (AMDV BPN5 Wild PL) as a representative. Among these seven sequences, four (57.14%, 95% CI = 18.41–90.10) were isolated from the American mink obtained from the NPN, while three (42.86%, 95% CI = 9.90–81.59) were isolated from the American mink obtained from the BPN. Two additional sequences (Acc. No.: OK263139—AMDV NPN4 Wild PL; OK263143—AMDV NPN8 Wild PL) also exhibited 100% homology (18.18%, 95% CI = 2.28–51.78), and we selected OK263139 (AMDV NPN4 Wild PL) for future analysis. In the group was one isolate with a unique sequence (Acc. No.: OK263140—AMDV NPN5 Wild PL). The isolates in the group showed slight differences, with an evolutionary divergence ranging from 0.33% to 1.32%. The substitutions in the *NS1* gene of our AMDV variants were compared to AMDV G (Acc. No. NC001662), as presented Figure 2. 

Amino acid (aa) substitutions in the NS1 protein of our AMDV variants were compared with the NS1 protein of AMDV G (Protein ID NP042872), as presented Figure 3. 

Details on the point mutations (transitions and transversions) that induce amino acid changes are presented Table 2.

The evolutionary relationships between the *NS1* gene variants obtained in this study and the other *NS1* genes of AMDV (Table 1) from NCBI are shown Figure 4. 

The isolates from the group were most similar to the isolates from first clade (group A) described previously by Kondracki et al. [24] and the sequences isolated in the USA—AMDV G (Acc. No. NC 001662); there is evolutionary divergence between the sequences of 5.80 and 6.86%, AMDV Utah (Acc. No. X77083), with an evolutionary divergence between sequences of 6.16 and 7.23%, and Germany, AMDV SL3 (Acc. No. X77085), with an evolutionary divergence between sequences of 6.16 and 7.23%. These variants also shared similarity with the nucleotide sequences isolated in Greece, AMDV isolate F8B5 F (Acc. No. MG366646.1), an evolutionary divergence between sequences of 7.23 and 7.59%, Latvia, AMDV A1 F (Acc. No. OR123963.1), an evolutionary divergence between sequences of 7.24%, and Poland, AMDV isolate ZP1b (I) F (Acc. No. MN175700.1), and an evolutionary divergence between sequences of 7.60%. The compared sequences were obtained from farmed mink.

The nucleotide sequence of the *NS1* AMDV gene of the AMDV NPN7 W Poland (Acc. No. OK263142.1) was different from those of the other variants isolated in this study, with an evolutionary divergence between sequences of 14.60 and 15.39%. The sequence of the AMDV NPN7 W Poland variant (Acc. No. OK263142.1) was the most similar to the sequences from second clad (group B) described previously by Kondracki et al. [24] and variants of AMDV isolated in Finland: AMDV HY 47 W (Acc. No. MG821256.1), AMDV HY 47 W (Acc. No. MG821256.1), AMDV FIN17/F40 F (Acc. No. MG821282.1). These had evolutionary divergences between sequences of 5.09%, 5.79%, and 7.60%, respectively. From Denmark, AMDV-United (Acc. No. X77085), an evolutionary divergence was found between sequences of 6.17%. From China, AMDV ADV-LN2 (Acc. No. GU183265.1), an evolutionary divergence was found between sequences of 6.87%. From Poland, AMDV mink PL 459 F (Acc. No. KY997051.1), AMDV PODK2a (IV) F (Acc. No. MN175729.1), AMDV WLKP2a (III) F (Acc. No. MN175724.1), and AMDV LUB1 (II) F (Acc. No. MN175735.1), an evolutionary divergence was found between sequences of 6.16%, 6.52%, 6.86%, and 6.89%, respectively. From Latvia, AMDV B4 F (Acc. No. OR123971.1), an evolutionary divergence was found between sequences of 7.22%. These *NS1* AMDV gene fragments were obtained from both farm-raised and free-ranging mink.

## 4. Discussion

The results of our study underscore a significant prevalence of AMDV in free-ranging American mink populations in the national parks in northeast Poland. The detection of the *NS1* gene in 50% of spleen samples from mink in Narew National Park (NNP) and 30% of spleen samples from mink in Biebrza National Park (BNP)—resulting in an overall prevalence of 42.31%—aligns with global reports of widespread AMDV presence in free-ranging mink populations; such studies have reported prevalence rates between 14.80% and 57.60% [3,4,8,23,60,61]. These findings corroborate earlier research by Jakubczak et al., who reported a 35% prevalence of AMDV in free-ranging mink from the same region in Poland in 2017 [43]. The detection of AMDV across different ecosystems suggests potentially serious ecological and health implications, particularly given the virus’s environmental persistence and high contagion [3,34,38].

Our phylogenetic analysis revealed that the majority of AMDV variants identified in this study (10 out of 11) belonged to a homogenous group, closely resembling clade 1 (group A), as described by Kowalczyk et al. and Kondracki et al. [20,24]. The variants exhibited an evolutionary divergence ranging from 0.33% to 1.32% and shared high genetic similarity with strains isolated in the USA, Germany, Greece, Latvia, and Poland [14,20,22,24,54,55]. The presence of a distinct variant (AMDV NPN7) further underscores the genetic diversity of AMDV within these populations, as it clustered with different clades described by Kowalczyk et al. and Kondracki et al. (clades II, III, and IV/group B) and with sequences from other European countries (Finland, Denmark, Latvia, Finland, Poland) and China [20,24,56,57,58,59].

The higher prevalence of AMDV found in samples from NNP compared to that found in samples from BNP suggests possible environmental or population density factors influencing virus transmission, although these differences were not statistically significant. This variation could also reflect different management practices across the parks, warranting further investigation.

Our findings, which identified similar AMDV variants in both farmed and free-ranging mink populations, suggest a potential interchange of viral strains between these groups, highlighting the risk of spill-back and spill-over events. Such interactions between domesticated and wild mink could contribute to the observed genetic diversity and have significant implications for AMDV transmission dynamics [3,38,45]. The presence of closely related AMDV strains in both free-ranging and farmed mink populations indicates that mink farms could be a source of AMDV transmission to wild populations, reinforcing the need for vigilant disease monitoring and management.

The One Health approach is particularly relevant in addressing the transmission of AMDV, as it emphasizes the interconnectedness of human, animal, and environmental health. Collaborative efforts among veterinary, wildlife, and public health sectors are essential in the effective surveillance, prevention, and control of AMDV spread. Continued research, including genetic analysis and epidemiological studies, is crucial for understanding the transmission dynamics of AMDV and implementing strategies to mitigate its impact on mink populations and broader ecosystems.

## 5. Conclusions

In conclusion, our study highlights the significant prevalence and genetic diversity of AMDV among free-ranging American mink in national parks in northeast Poland. The identification of closely related viral strains in both wild and farmed mink populations suggests potential cross-transmission between these groups, raising concerns about broader ecological and health impacts. The findings underscore the importance of continued surveillance, genetic analysis, and collaborative efforts across sectors to manage the spread of AMDV. Adopting a One Health approach is essential if we are to effectively address the interconnected challenges posed by this persistent and contagious virus.

## Figures and Tables

**Figure 1 animals-14-02584-f001:**
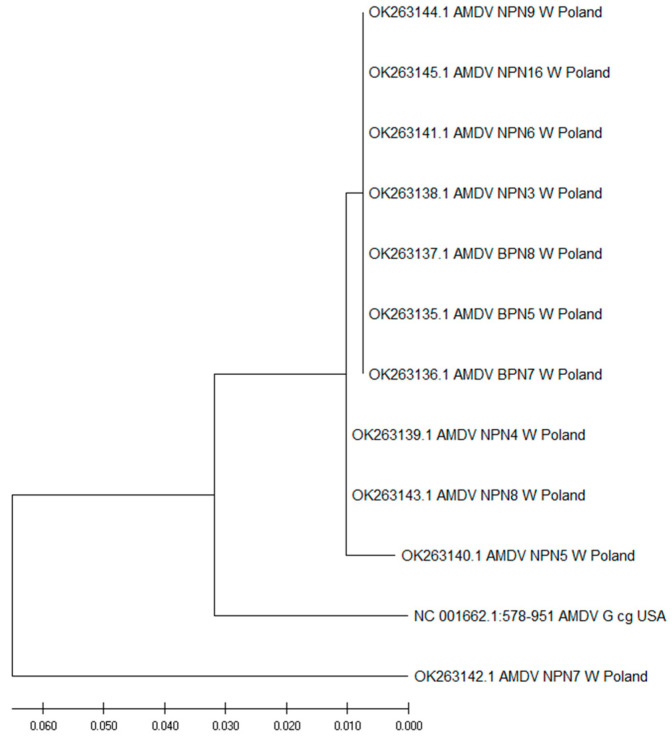
The evolutionary relationship between the *NS1* gene variants detected in this study and AMDV G (Acc. No. NC001662). Evolutionary history was inferred by using the maximum likelihood method and the Tamura–Nei model [54]. The tree with the highest log likelihood (−760.58) is shown. Initial tree(s) for the heuristic search were obtained automatically by applying the Neighbor–Join and BioNJ algorithms to a matrix of pairwise distances, as estimated using the Tamura–Nei model, and then by selecting the topology with superior log likelihood value. The tree was drawn to scale with branch lengths measured according to the number of substitutions per site. This analysis involved 12 nucleotide sequences. Codon positions included were as follows: 1st + 2nd + 3rd + noncoding. There was a total of 374 positions in the final dataset. Evolutionary analyses were conducted in MEGA X [52].

**Figure 2 animals-14-02584-f002:**
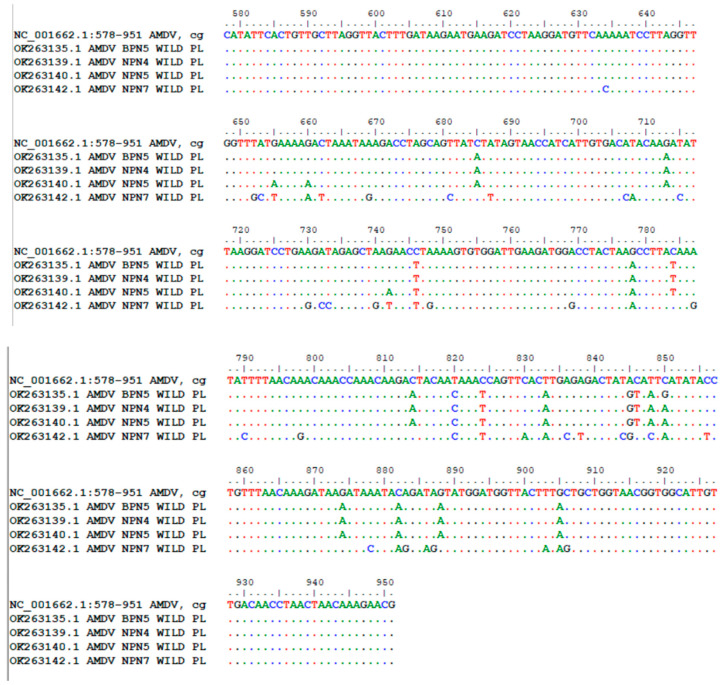
Nucleotide substitution of Polish isolates of *NS1* gene of AMDV in comparison to AMDV G genome (Acc. No. NC001662).

**Figure 3 animals-14-02584-f003:**
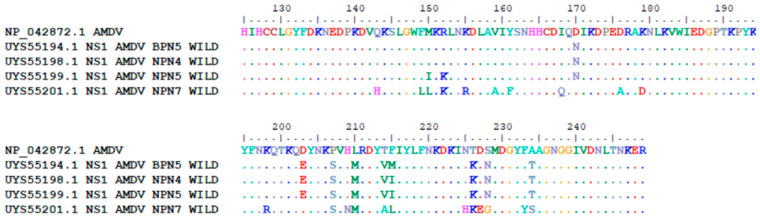
Amino acid (aa) substitutions in of Polish isolates on NS1 protein of AMDV in comparison to NS1 protein of AMDV G (Protein ID NP042872).

**Figure 4 animals-14-02584-f004:**
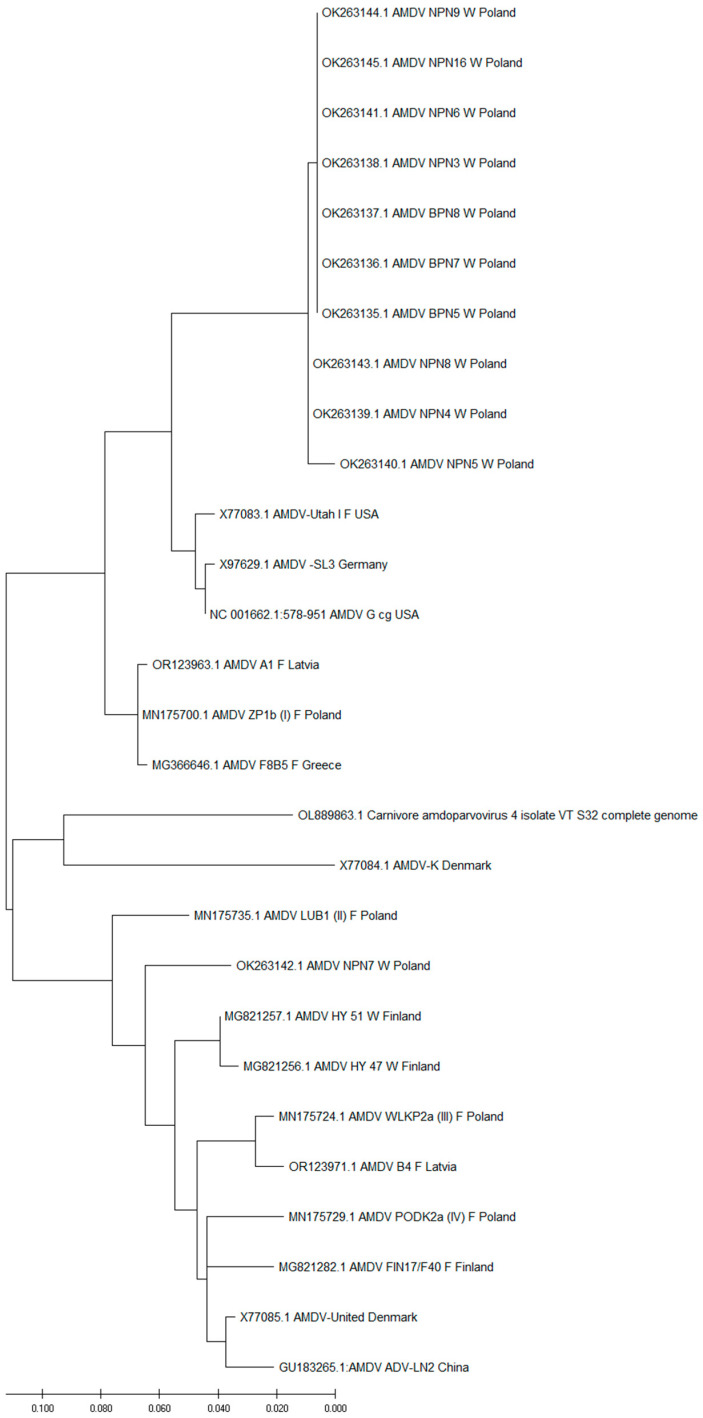
The evolutionary relationship between the *NS1* gene variants obtained in this study and the other *NS1* genes of AMDV (Table 1) from NCBI (redo analysis, including outgroup OL889863.1). The evolutionary history was inferred by using the maximum likelihood method and the Tamura–Nei model [54]. The tree with the highest log likelihood (−4694.52) is shown. Initial tree(s) for the heuristic search were obtained automatically by applying the Neighbor–Join and BioNJ algorithms to a matrix of pairwise distances estimated using the Tamura–Nei model, and then selecting the topology with the superior log likelihood value. The tree was drawn to scale with branch lengths measured according to the number of substitutions per site. This analysis involved 28 nucleotide sequences. Codon positions included were as follows: 1st + 2nd + 3rd + noncoding. Evolutionary analyses were conducted in MEGA X [52].

**Table 2 animals-14-02584-t002:** Point mutations that induce amino acid changes between isolates of *NS1* gene of AMDV in comparison to AMDV G (Acc. No. NC001662, position 578–951; protein ID NP042872, position 125–249).

Position in the *NS1* Sequence: Position in the NS1 Protein	AMDV G (NC001662) Nucleotides: Amino Acid	AMDV BPN5 W (OK263135) Nucleotides: Amino Acid	AMDV NPN4 W (OK263139) Nucleotides: Amino Acid	AMDV NPN5 W (OK263140) Nucleotides: Amino Acid	AMDV NPN7 W (OK263142) Nucleotides: Amino Acid
632–634: 143	CAA: Q	CAA: Q	CAA: Q	CAA: Q	CAC: H
650–652: 149	TTT: F	TTT: F	TTT: F	TTT: F	TTG: L
653–655: 150	ATG: M	ATG: M	ATG: M	ATA: I	CTT: L
659–661: 152	AGA: R	AGA: R	AGA: R	AAA: K	AAA: K
668–670: 155	AAA: K	AAA: K	AAA: K	AAA: K	AGA: R
680–682: 159	GTT: V	GTT: V	GTT: V	GTT: V	GCT: A
683–685: 160	ATC: I	**ATA: I**	**ATA: I**	**ATA: I**	ATC: I
686–688: 161	TAT: Y	TAT: Y	TAT: Y	TAT: Y	TTT: F
707–709: 168	ATA: I	ATA: I	ATA: I	ATA: I	CAA: Q
713–715: 170	GAT: D	AAT: N	AAT: N	AAT: N	**GAC: D**
728–730: 176	GAA: D	GAA: D	GAA: D	GAA: D	GAG: A
740–742: 179	AAG: K	AAG: K	AAG: K	**AAA: K**	GAT: D
797–799: 198	AAA: K	AAA: K	AAA: K	AAA: K	AGA: R
818–820: 203	AAT: D	AAC: E	AAC: E	AAC: E	AAT: D
824–826: 207	CCA: P	TCA: S	TCA: S	TCA: S	TCA: S
830–832: 209	CAC: H	CAC: H	CAC: H	CAC: H	AAC: N
833–835: 210	TTG: L	ATG: M	ATG: M	ATG: M	ATG: M
845–847: 214	ACA: T	GTA: V	GTA: V	GTA: V	GCA: A
848–850: 215	TTC: F	ATG: M	ATA: I	ATA: I	CTA: L
878–880: 225	AAT: N	AAT: N	AAT: N	AAT: N	CAT: H
881–883: 226	ACA: T	AAA: K	AAA: K	AAA: K	**AAG: K**
884–886: 227	GAT: D	GAT: D	GAT: D	GAT: D	GAA: E
887–889: 228	AGT: S	AAT: N	AAT: N	AAT: N	GGT: G
902–904: 233	TTT: F	TTT: F	TTT: F	TTT: F	TAT: Y
905–907: 234	GCT: A	ACT: T	ACT: T	ACT: T	AGT: S

Nucleotides: A—adenine; G—guanine; C—cytosine; T—thymine. Amino acids: A—alanine; D—aspartic acid; E—glutamic acid; F—phenylalanine; G—glycine; H—histidine; I—isoleucine; K—lysine; L—leucine; M—methionine; N—asparagine; Q—glutamine; R—arginine; S—serine; T—threonine; V—valine; Y—tyrosine. Silent mutation data is marked in bold text.

## Data Availability

Data are contained within the article.

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
