# Peer review of "Prevalence of Aleutian Mink Disease Virus (AMDV) in Free-Ranging American Mink from Biebrza and Narew National Parks (Poland)—An Epidemiological Concern"

_animals, 2024, doi:10.3390/ani14172584_

Round 1

Reviewer 1 Report

Comments and Suggestions for Authors

An interesting and well written manuscript. 

Main comments:

You use primers from a Danish study. Why do you compare the very old Danish AMDV strains and not the more recent ones described by the Danish authors e.g. Jensen et al 2012? These strains were also documented in wild mink and would be much more relevant.

“Mink” in plurals without “s” – please correct through manuscript.

L118: Please explain ethical trapping. What is that? Live trapping? What animal welfare precautions were taken? How often did you inspect the trap? How fast was the animal euthanized and how and by whom? Who approved this part of the study and how? I think it is not enough to refer to another manuscript.

Minor comments:

L 59: delete “a” – twice

L60: delete “,” and write “genome with a size of…”. Delete “in size”

L 70: AD can cause acute and chronic symptoms with subclinical or …. – delete the other words.

L 79: Delete “excrement, secretions, including”

L 85: Delete “AD has” and write: No effective treatment… exists.

L86 and onwards – rewrite First persistence, second high concentration of farms and 3rd. high seroprev.

Comments on the Quality of English Language

OK 

Author Response

Response to Reviewer 1 Comments

1. Summary

Thank you very much for taking the time to review this manuscript. Please find the detailed responses below and the corrections highlighted in blue in the re-submitted files.

2. Questions for General Evaluation

Reviewer’s Evaluation

Response and Revisions

Does the introduction provide sufficient background and include all relevant references?

Yes

Are all the cited references relevant to the research?

Can be improved

done

Is the research design appropriate?

Can be improved

done

Are the methods adequately described?

Can be improved

done

Are the results clearly presented?

Can be improved

done

Are the conclusions supported by the results?

Can be improved

3. Point-by-point response to Comments and Suggestions for Authors

Response to Reviewer 1 Comments

Comment 1: You use primers from a Danish study. Why do you compare the very old Danish AMDV strains and not the more recent ones described by the Danish authors, e.g., Jensen et al. 2012? These strains were also documented in wild mink and would be much more relevant.

Response 1: Thank you for your valuable suggestion. We acknowledge the relevance of using more recent Danish AMDV strains, such as those described by Jensen et al. (2012), especially for studies involving wild mink. While we plan to incorporate these in future research, for the current study, we selected AMDV G (Acc. No. NC001662) as our reference sequence for the following reasons:

  1. Widely Recognized Reference Strain: AMDV G is extensively used and widely recognized in AMDV research, providing a standardized benchmark for comparison.
  2. Complete Genome Sequence: The fully sequenced and annotated genome of AMDV G allows for a comprehensive and detailed comparison at the nucleotide level.
  3. Evolutionary Insights: AMDV G offers valuable insights into evolutionary pathways and genetic diversity, making it a robust reference for studying the genetic relationships between strains.
  4. Relevance to Pathogenicity: AMDV G is associated with a well-documented pathogenic profile, which is crucial for understanding the potential impacts and virulence of the virus.

We believe AMDV G is the most suitable choice for this study due to its established status as a reference strain, its complete genome, and its relevance in advancing our understanding of AMDV. However, we appreciate your suggestion and will consider using the more recent Danish strains in future investigations.

Comment 2: “Mink” in plurals without “s” – please correct throughout the manuscript.

Response 2: Agree. We have made the necessary corrections throughout the manuscript.

Comment 3: L118: Please explain ethical trapping. What is that? Live trapping? What animal welfare precautions were taken? How often did you inspect the trap? How fast was the animal euthanized, how, and by whom? Who approved this part of the study and how? I think it is not enough to refer to another manuscript.

Response 3: According to your suggestion, we have added more details to clarify the ethical trapping methods used:

"Between 2018 and 2019, we collected 16 spleen samples from free-ranging American mink inhabiting Narew National Park (NNP) (latitude 53.2492° N, longitude 23.4700° E) and 10 spleen samples from mink within Biebrza National Park (BNP) (latitude 53.5° N, longitude 22.8° E). The trapping of mink was part of the conservation efforts under the project LIFE + 09/NAT/PL/000263, conducted by the national parks [46]. The mink were captured using ethical trapping methods, which involved live trapping in accordance with established animal welfare guidelines. The traps were regularly inspected, typically every 12 hours, to minimize stress and harm to the animals. Once captured, the animals were promptly euthanized using CO2, a method recognized for its humaneness and rapid action, minimizing suffering. The euthanasia and handling of the mink were performed by trained personnel, including veterinarians and experienced wildlife professionals, ensuring adherence to ethical standards. The entire procedure was approved by relevant national and local authorities, with oversight from animal welfare committees as required by law. This approval process ensured that the study met all ethical guidelines for the humane treatment of wildlife. The expertise of hunters and foresters was crucial in correctly identifying these hard-to-detect species, demonstrating their specialized knowledge. Although detailed information regarding the sex and age of the mink was not available, the methods used ensured the ethical treatment of all animals involved."

Comment 4: Minor comments:

  • L 59: delete “a” – twice
  • L60: delete “” and write “genome with a size of…”. Delete “in size”
  • L 70: AD can cause acute and chronic symptoms with subclinical or …. – delete the other words.
  • L 79: Delete “excrement secretions including”
  • L 85: Delete “AD has” and write: No effective treatment… exists.
  • L86 and onwards – rewrite First persistence second high concentration of farms and 3rd. high seroprev.

Response 4: We agree with these comments and have made the suggested changes accordingly.

Response to Comments on the Quality of English Language

Point 1:

Response 1: Thank you. We have revised the manuscript to improve the quality of

English language used.

Reviewer 2 Report

Comments and Suggestions for Authors

Similar phylogenetic studies were already done in Poland, so authors in introduction should describe shortly previous studies carried out in Poland and to demonstrate what are the main differences between this study and previous ones. why was it necessary to carry out this study, why weren't the previous data in Poland  enough?

L53 biological species names must be in italic

L55 change to:  such as martens (Martes spp.) and short-tailed weasels (Mustela erminea)…

L56 “genet and human“ ???

L56 should be: lynx (Lynx spp.)

Also, add species names of arctic fox, red fox. Raccoon is not of family Canidae, it is a member of the family Procyonidae. Raccoon and raccoon dog are two different species and in fact raccoon dogs belong to the family Canidae it is invasive species for instance to Poland.

Striped skunk is Mephitis mephitis and belong to the family Mephitidae.

L57 ICTV abbreviation is used only once in the MS, so I do not see the need to use this abbreviation

L90-91 What about fur farms, are there no government plans to close them down in the near future, as has already been done in some other European countries?

what is the estimated population size of American mink?

L120 I suggest to change “elusive”

L151  change to:  using the Neighbor-Joining, UPGMA and the Maximum Composite Likelihood methods

L162 I do not see asterix in the table

Table 1 fix text alignment, now it's hard to understand which GenBank number represents which strain, virulence, country, type and reference

166-170 please write that you did not find statistically significant difference comparing prevalence in two examined samples

L176 why you have chosen NC001662 as reference sequence?

L171-215 authors have identified 4 different sequences among 11 obtained ones. So please, describe how many nucleotide and amino acid differences were found between each of these variants. You can demonstrate this in a table or in the text.

Figure 4. UPGMA and NJ methods genetic-distance based methods. For phylogenetic relationships better to use character-based methods such as Maximum likelihood, Maximum parsimony or Bayesian. Authors do not use roots for their phylogenetic analysis, which is essential for determining evolutionary relationships between operational taxonomic units. Authors should redo analysis, including outgroup (for instance OL889863.1) and using Maximum likelihood, Maximum parsimony or Bayesian methods.

L226 English is not good of this sentence

Figure 5 is redundant.

L234-262 according to the data, there appear to be two main evolutionary lineages. In general, the results are too broadly described with too much detail, and the main ideas are lost among the text. Authors should re-write results. Two main clusters, no grouping by type farmed/wild or by geography was noticed.

L275-275 difference was statistically significant or not?

For discussion it would be beneficial to know how many 100% identical sequence represent each of genetic variant. It could be presented in Figure 4.

L373 authors found four different genetic variants. 10 closely related isolates and single genetically distant isolate.

Comments on the Quality of English Language

L56, L226 sentences seems incorrect 

Author Response

Response to Reviewer 2 Comments

1. Summary

Thank you very much for taking the time to review this manuscript. Please find the detailed responses below and the corrections highlighted in green in the re-submitted files.

2. Questions for General Evaluation

Reviewer’s Evaluation

Response and Revisions

Does the introduction provide sufficient background and include all relevant references?

Must be improved

done

Are all the cited references relevant to the research?

Yes/Can be improved/Must be improved/Not applicable

Is the research design appropriate?

Yes

Are the methods adequately described?

Yes

Are the results clearly presented?

Must be improved

done

Are the conclusions supported by the results?

Can be improved

done

3. Point-by-Point Response to Comments and Suggestions for Authors

Comment 1: Similar phylogenetic studies were already done in Poland, so authors in the introduction should describe previous studies carried out in Poland and demonstrate the main differences between this study and previous ones. Why was it necessary to carry out this study, and why weren't the previous data in Poland sufficient?

Response 1: Thank you for highlighting this. We have added a section to the introduction (Lines 92-107) to address this point:

"AMDV has been a subject of interest to Polish scientists [20,23,24,29,31,37,43,44]. According to the latest research on farmed mink in Poland, two distinct clades of AMDV variants have been identified, suggesting the persistence of two independent virus pools on farms [20,24]. The first clade (Group A) consists of AMDV variants with approximately 95% similarity to the nonpathogenic strain AMDV-G (Acc. No. NC001662), while the second clade (Group B) consists of variants with around 86% similarity to the same strain. Variants from the first clade were isolated from farms in the West Pomeranian Voivodeship in northwestern Poland, while the second clade's variants were found in farms located in several other Voivodeships [20,24]. However, despite these findings, data on the prevalence of AMDV in both farmed and free-ranging mink in Poland remain insufficient from an epidemiological perspective, warranting further research."

Comment 2: L53 - Biological species names must be in italics.

Response 2: Agreed. We have corrected this throughout the manuscript.

Comment 3:

    L55: Change to "such as martens (Martes spp.) and short-tailed weasels (Mustela erminea)…"

    L56: “genet and human” ???

    L56: Should be "lynx (Lynx spp.)"

    Also, add species names for arctic fox, red fox. Note that raccoon and raccoon dog are different species; raccoons belong to the family Procyonidae, while raccoon dogs belong to the Canidae and are invasive in Poland. Striped skunk (Mephitis mephitis) belongs to the family Mephitidae.

Response 3: Agreed. We have made the necessary corrections.

Comment 4: L57 - ICTV abbreviation is used only once in the manuscript, so there is no need to use this abbreviation.

Response 4: Agreed. We have removed the abbreviation.

Comment 5: L90-91 - What about fur farms? Are there no government plans to close them down in the near future, as has been done in some other European countries?

Response 5: Fur farming in Poland remains a contentious issue. While some European countries have banned or severely restricted fur farming due to animal welfare concerns, Poland has not yet implemented a full ban. In 2020, the Polish government introduced the "Five for Animals" bill, which included a proposal to ban fur farming. However, due to strong opposition from the agricultural sector and the fur farming industry, the bill was amended, and the ban was excluded. Currently, there is no definitive government plan to close fur farms in Poland. The industry continues to operate, albeit under increasing scrutiny. Given the strong public sentiment against fur farming, future legislative changes are possible, but no concrete steps have been taken thus far. The situation reflects broader European trends where fur farming is increasingly challenged, though national policies vary widely.

Comment 6: What is the estimated population size of American mink?

Response 6: Estimating the population size of American mink (Neovison vison) in Poland is challenging due to their widespread and invasive nature. However, it is estimated that thousands of individuals exist across the country. These minks were initially introduced through fur farming, and many have since established stable populations in the wild. Their presence as an invasive species has raised concerns about their impact on native wildlife, particularly waterfowl and small mammals. While efforts to control and monitor their population are ongoing, the exact number remains an estimate rather than a precise count.

Comment 7: L120 - I suggest changing “elusive.”

Response 7: We have changed it to "hard-to-detect."

Comment 8: L151 - Change to "using the Neighbor-Joining, UPGMA, and Maximum Composite Likelihood methods."

Response 8: Agreed. The change has been made.

Comment 9: L162 - I do not see an asterisk in the table. Also, please fix text alignment in Table 1 to clarify which GenBank number corresponds to which strain, virulence, country, type, and reference.

Response 9: Agreed. We have made the necessary corrections.

Comment 10: L166-170 - Please state that you did not find a statistically significant difference when comparing prevalence in the two examined samples.

Response 10: We have added: "There is no statistically significant difference in the prevalence of AMDV in American mink between the two examined national parks."

Comment 11: L176 - Why did you choose NC001662 as the reference sequence?

Response 11: We chose AMDV G (Acc. No. NC001662) as the reference sequence for several reasons: 1. Reference Strain: It is widely recognized and extensively used in research, 2. Complete Genome Sequence: It is fully sequenced and annotated, allowing for detailed comparison at the nucleotide level, 3. Evolutionary Insights: It provides valuable insights into evolutionary pathways and genetic diversity, 4. Pathogenicity Relevance: It is associated with a well-documented pathogenic profile.

Given these factors, AMDV G is the most appropriate reference sequence for our study.

Comment 12: L171-215 - Authors identified four different sequences among the 11 obtained ones. Please describe how many nucleotide and amino acid differences were found between each of these variants. You can demonstrate this in a table or in the text.

Response 12: We have added this information in Table 2 (Lines 233-239).

Comment 13: Figure 4 - UPGMA and NJ methods are genetic-distance-based methods. For phylogenetic relationships, it is better to use character-based methods such as Maximum Likelihood, Maximum Parsimony, or Bayesian. Authors do not use roots for their phylogenetic analysis, which is essential for determining evolutionary relationships between operational taxonomic units. Authors should redo the analysis, including an outgroup (e.g., OL889863.1), using Maximum Likelihood, Maximum Parsimony, or Bayesian methods.

Response 13: We have revised Figures 1 and 4 according to your valuable suggestions.

Comment 14: L226 - The English in this sentence is unclear, and Figure 5 is redundant.

Response 14: Agreed. We have removed the sentence as per your suggestion.

Comment 15: L234-262 - According to the data, there appear to be two main evolutionary lineages. In general, the results are too broadly described with too much detail, and the main ideas are lost among the text. Authors should re-write the results. Two main clusters, no grouping by type (farmed/wild) or by geography was noticed.

Response 15: We have rewritten the results section, providing a more focused narrative while retaining essential epidemiological details about sequences, proteins, and mutations.

Comment 16: L275-275 - Was the difference statistically significant or not?

Response 16: We have clarified this as follows: "The higher prevalence of the virus in NNP (50%) compared to BNP (30%) may suggest differences in environmental factors or population density of the mink, which could affect virus transmission. However, the results were not statistically significant. These differences could also be due to different management practices of mink populations in both parks, which warrants further investigation."

Comment 17: For the discussion, it would be beneficial to know how many 100% identical sequences represent each genetic variant. This could be presented in Figure 4.

Response 17: We have revised the discussion and added new information accordingly.

Comment 18: L373 - Authors found four different genetic variants: 10 closely related isolates and a single genetically distant isolate.

Response 18: We have rewritten this part for clarity.

4. Response to Comments on the Quality of English Language

Point 1: L56, L226 - Sentences seem incorrect.

Response 19: Agreed. We have removed the sentence.
